# Physiological impact of nanoporous acupuncture needles: Laser Doppler perfusion imaging in healthy volunteers

**Koh-Woon Kim**[1☯], **Sanhwa Hong**[2,3☯], **Hong Soo Kim**[4☯], **Taehun Kim**[2], **Jaeha Ahn**[2], **Hyun-Seo Song**[2], **Yu-Kang Kim**[2,5], **Ju-Young Oh**[2,5], **Tae-Yeon Hwang**[2,5], **Hyangsook Lee**[2,5], **Su-Il In**[4]*, **Hi-Joon Park**[2,5]*

**1** Department of Korean Rehabilitation Medicine, College of Korean Medicine, Kyung Hee University, Dongdaemun-gu, Seoul, Republic of Korea, **2** Acupuncture and Meridian Science Research Center, Kyung Hee University, Dongdaemun-gu, Seoul, Republic of Korea, **3** Department of Meridian Medical Science, College of Korean Medicine, Graduate School, Dongdaemun-gu, Seoul, Republic of Korea, **4** Department of Energy Science and Engineering, DGIST, Hyeonpung-myeon, Dalseong-gun, Daegu, Republic of Korea, **5** Department of Anatomy and Information Science, College of Korean Medicine, Dongdaemun-gu, Seoul, Republic of Korea

☯ These authors contributed equally to this work.
* acufind@khu.ac.kr (HJP); insuil@dgist.ac.kr (SII)

**Data Availability Statement:** All relevant data are within the paper and its Supporting Information files.

## Abstract

### Background

Recently, porous acupuncture (PA), which is anodized to increase its surface area for higher stimulation intensity, was developed and showed significantly improved therapeutic effects with more comfort as compared with original acupuncture (OA) in vivo. However, the impact of PA on the change of local blood flow as well as its efficacy and acceptability has not yet been confirmed in a clinical trial. In a randomized, controlled crossover clinical trial, we investigated the effects of PA on the change in local blood flow using laser Doppler perfusion imaging and considered the sensation of pain intensity and discomfort severity using a visual analogue scale (VAS) to explore its physiological impact and the possibility of PA in clinical use.

### Methods

Twenty-one healthy participants were randomly treated with PA or OA on one side of Zusanli (ST36) and each participant served as his or her own control. Baseline local blood flow and galvanic skin response (GSR) were obtained for 5 min and acupuncture interventions were subsequently performed. Next, local blood flow and GSR were subsequently obtained for 10 min after insertion, 10 min after manipulation, and 5 min after the withdrawal of acupuncture. At the end of the experiment, participants were asked to indicate the sensation of pain intensity at each session of insertion, retention, manipulation, and withdrawal as well as the overall pain intensity and discomfort severity.

**Funding:** This research was supported by grants from the National Research Foundation of Korea, funded by the Korean government (2017R1A2B4009963), and the Korea Institute of Oriental Medicine (grant K18182) given to H.J.P. and by grants of the Korea Health Technology R&D Project through the Korea Health Industry Development Institute (KHIDI), funded by the Ministry of Health & Welfare (HI17C1357), given to S.I.I. The funders had no role in study design, data collection and analysis, decision to publish, or preparation of the manuscript.

**Competing interests:** The authors have declared that no competing interests exist.

## Results

PA significantly increased the local blood flow as compared with OA and there was no significant difference in GSR between patients treated with PA versus OA in each phase of insertion and manipulation. No significant difference in pain intensity or discomfort severity was found during manipulation, retention, or withdrawal of acupuncture.

## Conclusions

These results indicate that PA increases local blood flow, which can be closely related to the observed enhanced performance, without any associated discomfort or pain, suggesting its applicability in clinical practice.

## Introduction

Acupuncture, which has been practiced as a traditional medical procedure for more than twenty centuries [1], is now widely used in the treatment of many functional disorders such as dyspepsia [2, 3], musculoskeletal pain [4, 5], gynecological diseases [6, 7], and psychiatric disorders [8] and various kinds of chronic incurable diseases [9, 10]. To date, a considerable number of studies have been performed in the field of basic and translational research to reveal the mechanism underlying the clinical effects of acupuncture [11, 12]. While the anatomical structures of acupoints and the biological mechanism of acupuncture have not yet been clearly elucidated, previous studies provide general evidence that acupuncture produces a needling sensation via the mechanical stimulation of acupoints with the involvement of mechanoreceptors in the activated afferent nerve fibers [13–16]; subsequently, the release of endogenous opioids likely contributes to the therapeutic effects [17–20].

Focusing on the mechanical stimulation of acupuncture, various methodological procedures such as manual manipulation [21, 22], retention time [23], and stimulation frequency [24] as well as different forms of acupuncture like electroacupuncture [25], sticker acupuncture [26], and laser acupuncture [27] have been practically used and investigated. Needle factors such as material [28], diameter [29, 30], depth of insertion [29, 31], number of needles per session [32], and even needle surface [33, 34] have also been postulated to increase possible interaction between the inserted needle and mechanoreceptors in the surrounding tissue and finally affect the acupuncture efficacy [13, 14].

Recently, there has been considerable interest arising regarding developing a new class of acupuncture needles by modifying the needle surface in ways other than changing the needle diameter or insertion depth to improve stimulation intensity without meaningful discomfort. Kwon et al. [35] demonstrated that a coarse needle surface potentiated an analgesic effect elicited by acupuncture in rats with nociceptive pain, suggesting the significance of increasing friction and surface area for enhancing the effects of acupuncture needles. Accordingly, porous acupuncture (PA), which is anodized to increase its surface area to a value approximately 20 times greater than conventional acupuncture needles, was developed to increase the physical friction applied to the surrounding tissue for higher stimulation intensity [36]. PA presented significantly improved therapeutic effect with more comfort in alcohol-dependent [36] and colorectal cancer animal models [37].

However, there exists no evidence regarding the impact of PA on the change of local blood flow around acupoints, which might constitute a significant and useful indicator for

standardizing the stimulation intensity and assessing the peripheral effect of acupuncture. Various studies observing the physiological changes induced by acupuncture have been conducted and they reported that the change of local blood flow increased the experimental pressure threshold, which led to the enhanced therapeutic effect, e.g. analgesic effect of acupuncture, suggesting the significance of local blood flow as an objective parameter for acupuncture stimulation [24, 30, 38–40].

Moreover, the efficacy and acceptability of PA have not yet been proven in a clinical trial. As a pilot study, we previously collected 10 healthy participants, acupunctured on LI4 with PA performed in one side and ordinary acupuncture (OA) in the opposite, and measured local blood flow and galvanic skin response (GSR). Significantly increased local blood flow was observed in PA group as compared with OA group but there was no significant difference in the autonomic nervous system between PA and OA. Thus, we designed the present randomized, controlled crossover clinical trial to verify the effects of PA versus OA regarding the change in local blood flow. Subsequently, we investigated the difference of the sensation of pain intensity and discomfort severity between PA and OA to explore the possibility of PA in clinical use and the applicability in further research efforts.

## Methods

### Participants

Twenty-two healthy participants aged between 18 years and 50 years who were not vulnerable to and who did not have antipathy toward acupuncture were recruited using an advertisement. Volunteers with sensation-involving diseases such as paresthesia and anesthesia; local or general pain due to certain causes; hypotension; severe hypertension; bronchial diseases; heart diseases including arrhythmia, ischemic heart disease, and conduction defect; a past history of anaphylaxis or long-term use of steroid medications; who were taking medicines influencing autonomic nervous system; or who were pregnant or breastfeeding were excluded [39]. Detailed explanations of the experimental procedures that the recruits would be subjected to were given and the recruits provided informed consent for participation. This study was approved by the Kyung Hee University Ethics Committee (KHSIRB-17-045) and was conducted according to the principles of the Declaration of Helsinki [41].

### Experimental design and procedures

This study's design was a randomized crossover in which each participant served as his or her own control. Participants were divided into four groups in random order and session orders were determined with a random number table generated by Excel (Microsoft, Redmond, WA, USA). Each experiment was separated by 30 minutes. Among the four groups, one group received PA on the left leg and, after 30 minutes of rest, received OA on the right leg, while another group first received PA on the right and then OA on the left. In the same manner, the other two groups respectively received either OA on the left followed by PA on the right or OA on the right followed by PA on the left (Fig 1a). Tests were carried out between August 3 and 31, 2017.

The procedures in this study were conducted in a light-conditioned and quiet room, in which the temperature, humidity, and brightness were respectively maintained at 25°C ± 1°C (mean ± standard deviation), 40%, and 40 Lux. Demographic data including gender; age; height and weight; blood pressure and pulse rate; body temperature; past history of and any present illnesses; and history of alcohol ingestion, smoking, and medication were collected from all participants. All participants refrained from consuming alcohol, caffeine, or medication for 12 hours before the experiment. Additional baseline characteristics such as sleeping

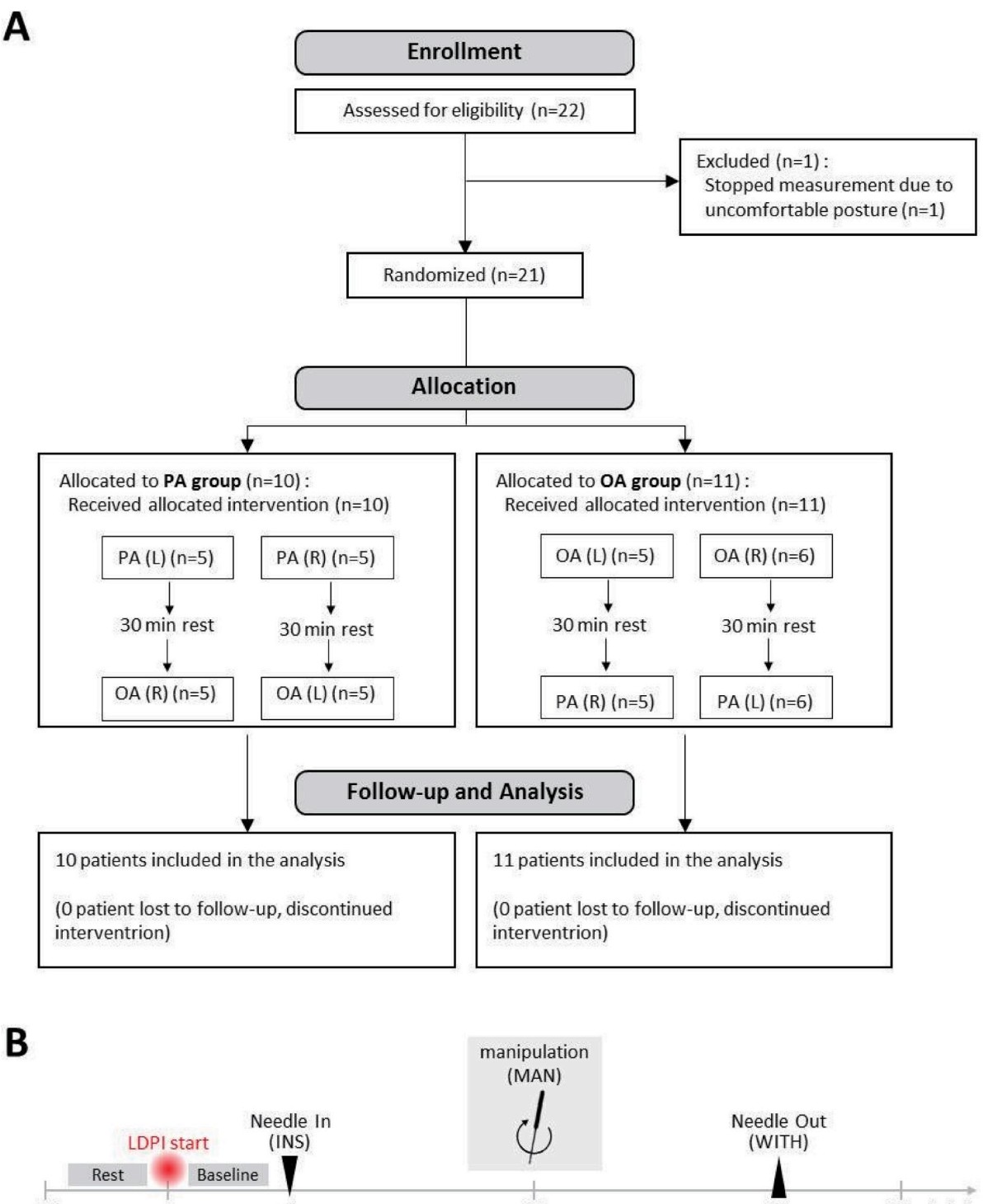

**Fig 1. The schematic diagram of the experiment.** Study flow (**a**). After recruitment, participants were randomly allocated into four groups. e.g., five participants were acupunctured with PA on the left side first and, after 30 minutes, with OA on the right side. Flow of each session (**b**). Participants received an explanation of this experiment, signed the consent form, and filled out the case report form. After a five-minute rest, laser Doppler perfusion imaging and GSR were set and measured for 30 minutes: five minutes for baseline, 10 minutes after the insertion of acupuncture, 10 minutes after manipulation at 2 Hz for 30 seconds, and five minutes after the withdrawal of acupuncture. Manipulation techniques were performed as tonifying and reducing, which involved bidirectional rotation. OA, ordinary acupuncture; PA, porous acupuncture; L, left; R, right; INS, insertion; MAN, manipulation; WITH, withdrawal.

**Fig 2. Practical experiment photo.** Laser Doppler perfusion imaging was focused on the subject's left ST36 acupoint (**a**). Retention of acupuncture needles of PA (**b**) and OA (**c**). Laser Doppler perfusion imaging was estimated within the square region (3.5 × 3.5 cm). Manual rotation of acupuncture needles of PA (**d**) and OA (**e**). OA, ordinary acupuncture; PA, porous acupuncture.

hours of the day before the experimental day, use of alcohol or caffeine with the time of latest use, and previous experiences of acupuncture treatments along with times of treatments and their effects were also gathered as further information. Considering that the expression of pain sensation involves a subjective element and, therefore, is usually easily affected by individual expectancy, character or personality, and other psychological factors, participants were asked to answer additional questionnaires including the Beck Depression Inventory (score range: 0–63 points; a higher score means more depressive; participants were excluded if their score was 10 points or more), the Acupuncture Expectancy Scale (AES) [42] (range: 4–20 points; higher score means higher expectancy), and the Acupuncture Fear Scale (AFS) [43] (range: 16–80 points; higher score means more fear).

After five minutes of resting, participants were laid down in a supine position with their legs stretched and the hip joint medially rotated. Baseline local blood flow and GSR were obtained for five minutes and acupuncture interventions were subsequently performed. Next, local blood flow and GSR were subsequently obtained for 10 minutes after insertion, 10 minutes after manipulation, and five minutes after the withdrawal of acupuncture (Fig 1b). A practical experiment photo is shown in Fig 2. Finally, at the end of the experiment, participants were asked to indicate the sensation of pain intensity at each session of insertion, retention, manipulation, and withdrawal as well as the overall pain intensity and discomfort severity using the visual analogue scale (VAS). Needling sensation at each session was also assessed using the Acupuncture Perception Questionnaire (APQ) (S1 Table).

## Porous acupuncture and original acupuncture treatment

Both PA and OA stimulation were performed on one side's leg of the participant at ST36 (on the anterior aspect of the lower leg, 9.9 cm below ST35, middle finger from the anterior crest of the tibia), a representative acupoint which is clinically frequently used. Prior to baseline, the skin was cleaned with alcohol at the acupoint in question. Both PA and OA needles (0.30 mm in diameter, 6 cm in length; DongBang Acupuncture Inc., Boryeoung, Republic of Korea) were inserted to 10 mm in depth, controlled by an empty guide tube, and, after 10 minutes of retention, manipulated for 30 seconds. Manipulation techniques were performed as tonifying and reducing, which involved bidirectional rotation (2 Hz). A previous study [24, 44] showed repeated manipulation improves acupuncture effects and, traditionally, manual acupuncture uses manipulation to reinforce its effects [45]. The same doctor of Korean medicine having seven years of clinical experience with a certified license performed all acupuncture treatments.

The PA needles were prepared by electrochemical anodization of the OA needles, which were conventional stainless steel acupuncture needles (0.30 mm in diameter, 6 cm in length; DongBang Acupuncture Inc., Boryeoung, Republic of Korea), as reported previously. The PA

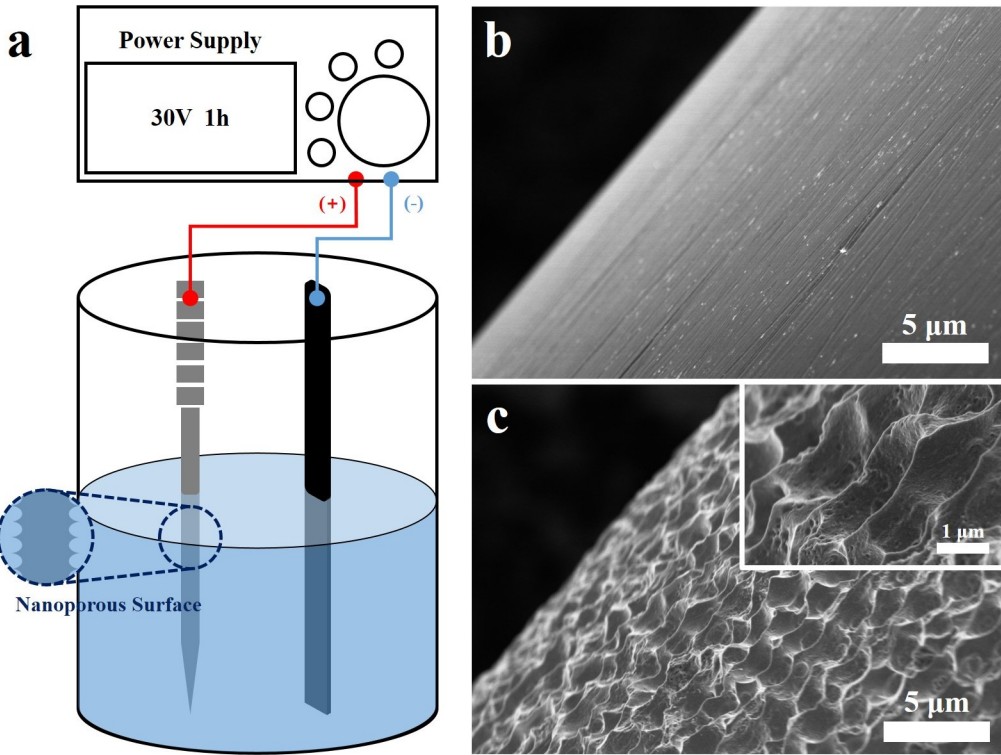

**Fig 3. Surface of PA and OA by high resolution scanning electron microscope.** Schematic diagram showing the electrochemical anodization step of PA (**a**). Surface of OA (**b**). Normal and magnified surface of PA (**c**). Scale bar represents the 5 μm and 1 μm each. OA, ordinary acupuncture; PA, porous acupuncture.

was fabricated using an anodization voltage of 30 V, and showed an estimated surface area of 0.0328 $m^2 \cdot g^{-1}$, a maximum value approximately 20 times higher than OA (0.0058 $m^2 \cdot g^{-1}$) (Fig 3) [36, 37]. It was approved as a medical instrument by the Korean Ministry of Food and Drug Safety after safety and toxicity testing on May 10, 2017.

## Local blood flow using laser Doppler perfusion imaging

A PIM3 system (Perimed AB, Järfälla, Sweden) was used to measure local blood flow by perfusion unit in this study. The area of measurement was 3.5 cm × 3.5 cm around the acupuncture stimulus site of ST36 on the leg. The distance between the detector and the tissue was fixed to 40 cm with normal resolution (Fig 2). To minimize movement, the ankle was fixed with a kapok-filled vacuum cushion. Local blood flow was measured for a total of 30 minutes, including for five minutes at baseline, 10 minutes after insertion, 10 minutes after manipulation and five minutes after the withdrawal of acupuncture. In analysis, the region of interest was 1.0 cm × 1.0 cm around the acupoint of ST36.

## Pain intensity and discomfort severity using the visual analogue scale

The sensation of pain intensity at each session of insertion, retention, manipulation, and withdrawal along with the overall pain intensity and discomfort severity were assessed using the VAS (range 0–10 points; higher score means more pain and discomfort) at the end of the experiment.

### Needling sensation using the acupuncture perception questionnaire

Needling sensation at each session of insertion, retention, manipulation, and withdrawal was assessed using the APQ (S1 Table) at the end of the experiment.

### Galvanic skin response

The degree of local sweating, or GSR, was measured with a Biopac MP150 system (Biopac Systems Inc., Goleta, CA, USA). The change of electrical impedance was obtained by two electrodes attached to participants' index and middle fingers on the right side with stabilization ensured in the supine position to exclude the influence of external environment. GSR was measured for a total of 30 minutes, including for five minutes at baseline, 10 minutes after insertion, 10 minutes after manipulation, and five minutes after the withdrawal of acupuncture. Participants were advised to concentrate on the needling sensation without sleeping, talking, or moving during the experiment procedures.

### Blinding

The practitioner was aware of the allocation arm while the participants, outcome assessors, and statisticians performing the data analysis were blinded to the treatment allocation. At the end of the study, participants were asked to choose one statement ("I felt little difference," "I felt some difference," or "I felt definite difference") to assess whether they felt a difference between the acupuncture treatments subsequently applied to each side.

### Adverse events

The adverse events (AEs) known related to acupuncture treatment include local bleeding or pain at the acupuncture points, local redness or bruising, itching, and dizziness during treatment [46]. According to the planned experimental protocol, any expected or unexpected AEs within a week from the experiment were recorded in detail including the date of occurrence, severity, causal relationship with the treatment, action or treatment, and progress.

### Statistical analysis

Statistical analyses were carried out using the Statistical Package for the Social Sciences version 23.0 software (IBM Corp., Armonk, NY, USA). Unless otherwise stated, data are presented as the mean ± standard deviation for continuous variables and the frequency and percentage (%) for dichotomous or categorical variables. $P$-values $< 0.05$ were considered to be statistically significant.

In the analysis of local blood flow and GSR, we used the rate of changes from baseline considering the individual differences of baseline local blood flow and GSR. The rate of change was calculated using the formula: (change from baseline)/baseline × 100.

Analyses were performed after checking all parameters were normally distributed using the Shapiro–Wilk one-sample test. Differences within or between groups were evaluated using a paired-sample $t$-test or Wilcoxon signed-rank test.

## Results

Twenty-two healthy participants (18 men and four women) were recruited and, finally, 21 participants were included for analysis except a man who was excluded due to posture discomfort (Fig 1a). Baseline characteristics and demographic data of participants are presented in Table 1.

**Table 1. Demographic data of participants.**

| Variable | Value (mean ± SD) |
|---|---|
| Gender (number, %) | Male (17, 81.0), Female (4, 19.0) |
| Ages, years | 36.9 ± 9.2 |
| Height, cm | 171.0 ± 7.8 |
| Weight, kg | 66.7 ± 11.9 |
| Body mass index (BMI) | 22.7 ± 2.9 |
| Beck Depression Inventory (BDI, 0–63) | 4.0 ± 4.4 |
| Acupuncture Expectancy Scale (AES, 5–20) | 14.7 ± 4.0 |
| Acupuncture Fear Scale (AFS, 16–64) | 24.8 ± 9.1 |

SD, standard deviation.

## Changes in local blood flow

The changes of skin blood perfusion over time are depicted in Fig 4. Within each group, skin blood perfusion after insertion, manipulation, and withdrawal was significantly increased from baseline in both the OA ($p < 0.05$ each) and PA ($p < 0.01$ each) groups. Between groups, there was no significant difference from baseline to three minutes after insertion, while subsequent skin blood perfusion were significantly different from four minutes after insertion to manipulation ($p < 0.05$), during one minute after manipulation ($p = 0.002$), and from one minute after manipulation to withdrawal and even after withdrawal ($p < 0.001$).

The changes in mean skin blood perfusion and the exemplary blood perfusion imaging by session classified as baseline, insertion (from insertion to manipulation), manipulation (from manipulation to withdrawal), and withdrawal (after withdrawal) are presented in Fig 5. The increase rates of mean skin blood perfusion as compared with baseline during each session of insertion, manipulation, and withdrawal were all statistically significant in both the OA (113.0%, 118.8%, and 111.9% respectively; $p < 0.05$ each) and PA (131.9%, 200.6%, and 184.5% respectively; $p < 0.01$ each) groups. In the session of manipulation versus insertion,

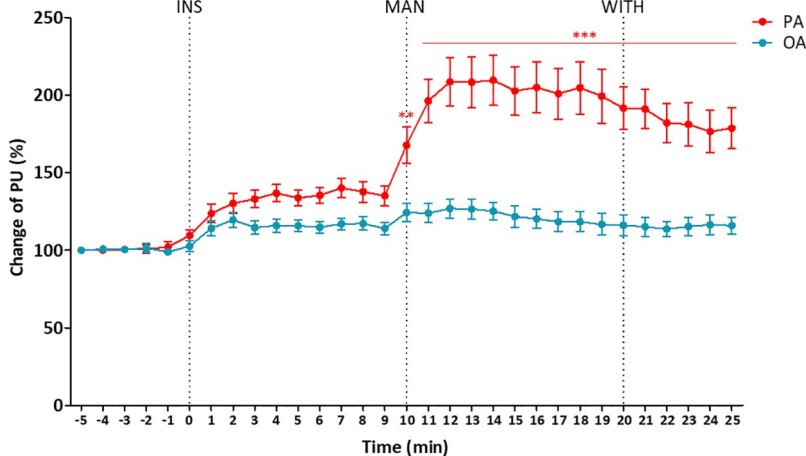

**Fig 4. Changes of skin blood perfusion over time.** The changes of PU from baseline per minute were compared by paired t-test. ** $p < 0.01$, and *** $p < 0.001$ between PA and OA. Values presented as means ± standard deviations. OA, ordinary acupuncture; PA, porous acupuncture; PU, perfusion unit; INS, insertion; MAN, manipulation; WITH, withdrawal.

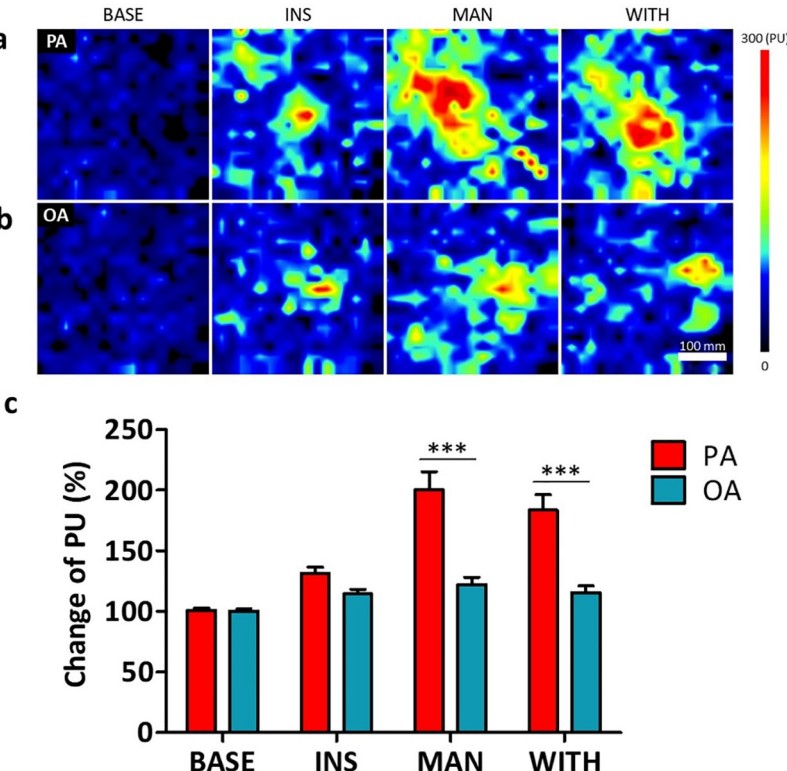

**Fig 5. The exemplary blood perfusion imaging in each session (Subject #7) and the changes of skin blood perfusion by session.** Blood skin perfusion imaging in each session of PA (**a**) and OA (**b**). The bar on the right represents the blood perfusion unit by color bar. The scale bar below = 350 mm. The mean changes of perfusion unit from baseline for each session were compared by paired t-test (**c**). *** $p < 0.001$ between PA and OA. Values presented as means ± standard deviations. OA, ordinary acupuncture; PA, porous acupuncture; PU, perfusion unit; BASE, baseline; INS, insertion; MAN, manipulation; WITH, withdrawal.

the mean skin blood perfusion was significantly increased with PA (52.1%; $p < 0.01$), while OA showed no significant increase (5.2%). Between groups, the PA group showed significant increases as compared with the OA group at each session of insertion, manipulation, and withdrawal ($p < 0.001$ each).

The changes of mean skin blood perfusion for each session according to the order or acupunctured side are depicted in S1 Fig. There were no significant differences between left and right (S1a and S1b Fig) and between first and second treatment (S1c and S1d Fig and S2 Table) in both the OA and PA groups.

## Pain intensity and discomfort severity

The changes of pain intensity using the VAS by session classified as insertion, retention, manipulation, and withdrawal (after withdrawal) are presented in Fig 6. The subsequent VAS scores at each session were 1.4 ± 1.6, 1.4 ± 2.1, 3.7 ± 2.7, and 0.3 ± 0.5, respectively, in the OA group and 2.1 ± 2.0, 1.1 ± 2.1, 4.2 ± 2.7, and 0.5 ± 1.4, respectively, in the PA group. Except for the insertion session, which showed a significant difference ($p < 0.05$), there were no significant differences noted between groups during each session of retention, manipulation, and withdrawal.

Separately assessed overall VAS pain ratings during the whole 30-minute experiment of the OA (2.0 ± 2.4) and PA (2.8 ± 2.3) groups showed no significant difference (Fig 6 and Table 2),

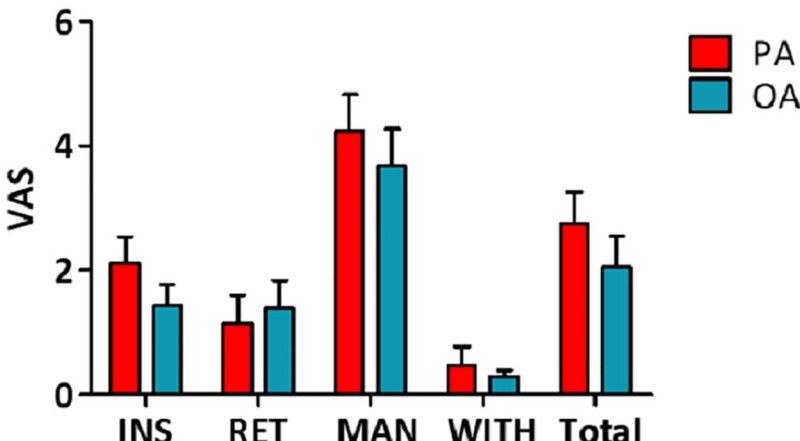

**Fig 6. The changes of pain ratings by session.** The differences of VAS pain intensity in each session and in the overall session. Values presented as means ± standard deviations. OA, ordinary acupuncture; PA, porous acupuncture; PU, perfusion unit; VAS, visual analogue scale; INS, insertion; RET, retention; MAN, manipulation; WITH, withdrawal.

as the overall VAS discomfort severity scores of the OA (2.4 ± 1.4) and PA (2.9 ± 1.5) groups also demonstrated no significant difference (Table 3). For further analysis, the participants were classified into subgroups according to their pain ratings (Table 2) and discomfort scores (Table 3).

**Table 2. Pain assessment between PA and OA groups.**

| Pain | n (%) | VAS (mean ± SD) | | |
|---|---|---|---|---|
| | | PA | OA | *p*-value |
| Total | 21 (100) | 2.8 ± 2.3 | 2.0 ± 2.4 | 0.3239 |
| PA > OA | 9 (42.9) | 4.0 ± 2.3 | 1.3 ± 1.5 | |
| OA > PA | 2 (9.5) | 3.0 ± 2.8 | 7.5 ± 0.7 | |
| No difference | 10 (47.6) | 1.6 ± 1.7 | 1.6 ± 1.7 | |

PA > OA means PA was more painful than OA; OA > PA means OA was more painful than PA; No difference means the level of pain was similar in both. OA, ordinary acupuncture; PA, porous acupuncture; VAS, visual analogue scale; SD, standard deviation.

**Table 3. Discomfort assessment between PA and OA groups.**

| Discomfort | n(%) | VAS (mean ± SD) | | |
|---|---|---|---|---|
| | | PA | OA | *p*-value |
| Total | 21 (100) | 2.9 ± 1.5 | 2.4 ± 1.4 | 0.2904 |
| PA > OA | 10 (47.6) | 4.0 ± 1.2 | 1.5 ± 0.8 | |
| OA > PA | 7 (33.3) | 1.7 ± 0.8 | 3.8 ± 1.1 | |
| No difference | 4 (19.1) | 2.3 ± 1.0 | 2.3 ± 1.0 | |

PA > OA means PA was felt more uncomfortable than OA. OA > PA means OA was felt more uncomfortable than PA. No difference means the level of discomfort was similar to each other. OA, ordinary Acupuncture; PA, porous acupuncture; VAS, visual analogue scale; SD, standard deviation.

## Needling sensation

Subjective reports regarding the needling sensation of acupuncture in the PA and OA groups at each session of insertion, retention, manipulation, and withdrawal assessed using the APQ (S1 Table) are shown in Table 4. The sums of the numbers of responders for the various needling sensations reported were not significantly different between groups at each session, while the sensation of dullness was reported more frequently during the insertion and manipulation sessions in the PA group than in the OA group (PA vs. OA: n = 4 vs. n = 1 for insertion and n = 9 vs. n = 4 for manipulation).

## Galvanic skin response

The changes of GSR, which was measured to assess the influence of PA and OA on the autonomic nervous system, are depicted in Fig 7. There were no significant differences in GSR within the PA and OA groups for each phase of insertion and manipulation. However, GSR gradually increased in the PA group after the manipulation of acupuncture, while GSR was constantly decreased in the OA group, though both of these findings without statistical significance. As a result, GSR showed a statistical difference between groups after the withdrawal of acupuncture according to such a tendency.

## Blinding

At the end of the study, to assess whether they felt a difference between PA and OA, participants were asked to choose one statement from among the following: "I felt little difference," "I felt some difference," and "I felt a definite difference." The subsequent responder numbers

**Table 4. Subjective reports on the sensation of acupuncture in PA and OA groups.**

|  | Insertion | | Retention | | Manipulation | | Withdrawal | |
|---|---|---|---|---|---|---|---|---|
|  | PA | OA | PA | OA | PA | OA | PA | OA |
| No sense | 6 | 4 | 10 | 8 | 1 | 2 | 13 | 15 |
| Pricking | 4 | 4 | 1 |  | 6 | 4 | 2 |  |
| Compressing or pressuring | 2 | 4 | 1 | 1 | 1 |  |  |  |
| Refreshing/relieving | 2 | 2 |  | 1 | 1 | 2 | 1 | 3 |
| Spreading out |  | 2 | 4 | 4 | 1 | 1 |  |  |
| Relieving sensation of tense or tight muscles |  |  |  |  |  | 1 |  |  |
| Activated blood circulation |  |  | 3 | 3 | 1 | 1 | 1 |  |
| Dull | 4 | 1 | 1 | 2 | 9 | 4 |  |  |
| Tickling |  |  |  |  | 2 | 2 |  |  |
| Achy | 1 |  |  |  | 2 | 2 |  |  |
| Surging/opening flow of stuffed or choked feeling |  |  |  |  |  | 1 | 3 | 1 |
| Gentle/soft touch |  | 1 |  |  |  |  |  | 2 |
| Shooting | 1 | 1 |  |  | 1 | 1 |  | 1 |
| Numb |  |  |  | 1 | 1 | 1 |  |  |
| Burning |  |  |  |  |  | 1 |  |  |
| Heavy |  | 1 | 1 | 2 | 1 | 1 |  |  |
| The number of sensations reported | 14 | 16 | 11 | 14 | 26 | 22 | 7 | 7 |

Numbers refer to the number of the responders. The number of sensations reported refers to the sum of the responders excluding the answer of "no sense." OA, ordinary acupuncture; PA, porous acupuncture.

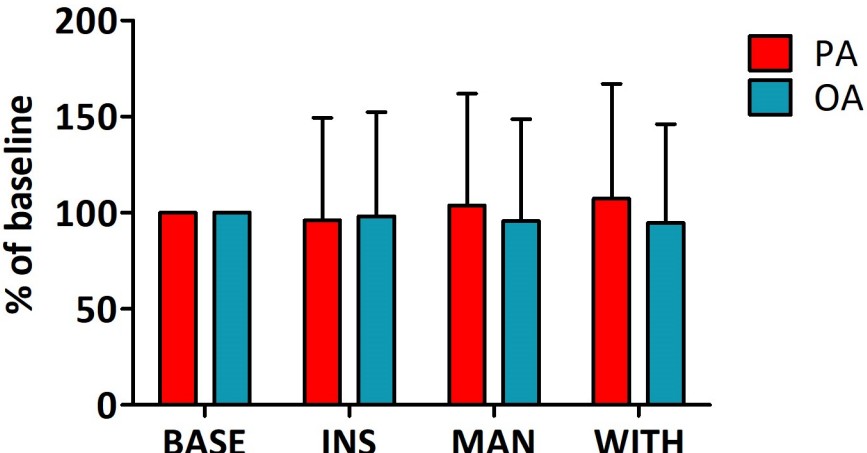

**Fig 7. Changes of GSR in the PA and OA groups.** The GSR was measured at baseline (five minutes) and after insertion (10 minutes), manipulation (10 minutes), and withdrawal (five minutes) of the PA or OA needle. Values presented as means ± standard deviations. OA, ordinary acupuncture; PA, porous acupuncture; BASE, baseline; INS, insertion; MAN, manipulation; WITH, withdrawal.

for each answer were 12 (57.1%), six (28.6%), and three (14.3%) respectively, which revealed that 85.7% of participants could not feel a definite difference between PA and OA.

### Adverse events

Any expected or unexpected AEs were not reported within a week after the experiment.

### Discussion

In this randomized, controlled crossover clinical study, we sought to investigate the effects of PA, focusing on the change of local blood flow as an objective parameter for acupuncture stimulation, which is supposed to be one of the physiological changes induced by acupuncture [24, 38–40]. PA significantly increased the local blood flow as compared with OA, especially in the phase of manipulation. The possibility of an unexpected influence by the order or acupunctured side was excluded, as there were no significant differences between the left and right sides and between the first and second treatments in the OA and PA groups. To exclude the possible influence on the answer to the blinding question by the crossover design, we chose the options for blinding question focusing on the difference between the first and second acupuncture treatments: "I felt little difference," "I felt some difference," and "I felt a definite difference" rather than "the former was PA and the latter was OA," "the former was OA and the latter was PA," and "I don't know". Furthermore, the probable individual difference of the autonomic nervous system (ANS), which could also affect blood flow changes, was simultaneously investigated to show no significant difference between the OA and PA groups in terms of the GSR, increasing the reliability of the study results. The results suggest that PA, with only the needling characteristics being different from OA, has a greater impact on local blood flow change versus OA.

There are several possible reasons or mechanisms to explain why PA had a different physiological influence on local blood flow. First, the higher stimulation intensity to the surrounding tissue induced by the increased physical friction and surface area of PA might have elicited a more significant increase of local blood flow. A previous study [24] demonstrated that, the higher the stimulation intensity to the connective tissue, the more increased the peripheral

blood flow was. Langevin et al. [47, 48] suggested connective tissue winding as the mechanism responsible for the increase in pullout force induced by needle rotation or needle manipulation and the key to acupuncture's therapeutic mechanism. Our study results indicating an approximately 200% increased rate in mean skin blood perfusion versus at baseline in PA group, which was particularly high during the session of manipulation, support the hypothesis of connective tissue involvement. Second, the higher neuronal activity induced by increased mechanical stimulation to the receptors in nerve fibers is likely to be related, considering that a previous study [31] suggested a greater number of receptors was the mechanism responsible for increased stimulation intensity and a greater efficiency. In et al. [36] found that PA showed a significant enhancement in neuronal activity as compared with conventional acupuncture; still, however, further research on the relation of neuronal activity or nerve conduction and the local blood flow change is necessary. Third, physiological active substances such as vasoactive substances released by noxious stimulation of PA might be candidates for the reason of increased blood flow. In previous studies [49–52], there was a relationship between blood flow increases and the intensity of the stimuli, while the release of vasodilative substances, such as calcitonin gene–related neuropeptide (CGRP), was suggested to be involved in the mechanisms of blood flow increase through the consequences of axon or dorsal root reflexes [53]. Sato et al. [52] concluded that antidromic vasodilatation in skeletal muscle for three to 15 minutes following 30 seconds of stimulation of afferent fibers in the dorsal roots is mediated by the release of CGRP from afferent nerve terminals. In this study, the acupoint of ST36 on the anterior aspect of the lower leg, which is anatomically adjacent to the anterior tibial artery and vein, was selected and acupuncture needles were inserted to 10 mm in depth after 10 minutes of retention and manipulated for 30 seconds to induce enough needling intensity to the anterior tibial muscle.

Other possible factors, which could additionally affect the blood flow changes, also must be thought of. First, interactions with sympathetic neurons of the needling must be considered in the mechanisms of blood flow changes [49, 52, 54–57]. In this study, though there were no significant differences in the GSR within the PA and OA groups in each phase of insertion and manipulation, the GSR gradually increased in the PA group following the manipulation of acupuncture but was constantly decreased in the OA group, albeit without statistical significance in both cases. As a result, the GSR showed a statistical difference between groups after the withdrawal of acupuncture according to such a tendency. However, considering that the effects of sympathetic neurons on blood flow are triggered quickly and are suggested to outlast the stimulation by a few seconds only, such effects are likely to be hidden by the more powerful effects of vasodilative substances induced by the needling [49, 55, 57]. Second, anxiety prior to the intervention or a sense of familiarity to the situation might have influenced the effects of needling on blood flow in various ways [58, 59]. In this study, the AES [42], the AFS [43], and previous experiences of acupuncture treatments were assessed as subjective elements prior to the experiment to exclude the influences of psychological factors on the effects of acupuncture and blood flow change. Third, the anatomical characteristics of acupoints could affect the blood flow change. The acupoint of ST36 selected in this study is anatomically adjacent to the anterior tibial artery and vein involving the anterior tibial muscle, which can be directly affected by blood vessels and skeletal muscle. However, considering the results of our pilot study previously mentioned, which showed a similar pattern to that in this study regarding local blood flow change with acupuncture occurring on LI4 (on the back of the hand) with PA in one side and OA in the other, the influence of the acupoint location might be less significant than that of the needling characteristics of PA.

To explore the possibility of PA in clinical use and its applicability in further investigations, we subsequently evaluated the differences in the sensations of pain intensity and discomfort

severity between PA and OA. Except for regarding the insertion session, there were no significant differences in the changes of VAS pain intensity between the groups at each session of retention, manipulation, and withdrawal. Separately assessed overall VAS pain ratings and discomfort severity scores during the whole 30-minute experiment of the OA and PA groups showed no significant differences. In the subgroup analysis performed according to the pain ratings and the discomfort scores, it was shown that the number of participants who answered, "OA was more painful or bothersome than PA" or "OA and PA were similar" was more than the number of those who answered "PA was more painful or bothersome than OA." In addition, for the subjective reports given on the needling sensation, the sums of the numbers of responders for the various needling sensations reported were not significantly different between the groups at each session, while the sensation of dullness was reported more frequently in the insertion and manipulation sessions in the PA group than in the OA group, a result that might be partly induced by the difference of the needle surface winding up the connective tissue and muscle. Though the individual difference of pain sensitivity should be considered to generalize these results, they suggest the applicability of PA in clinical practice.

Still, importantly, this study had several limitations. First, its sample size was too small and so should be regarded only as a pilot study, with further investigations with larger sample sizes powered to find the change that occurred in this study required to draw definite conclusions. Second, as we adopted a crossover design to reduce the influence of confounding covariates and for statistical efficiency with fewer subjects, problems related with the issues of "order" effects and "carry-over" between treatments have arisen, though the possibility of an unexpected influence by the order was excluded as there were no significant differences between the first and second treatments in the OA and PA groups. Third, participants often complained of discomfort from the posture held during measuring of local blood flow, which might have influenced the blood flow change. Finally, our study was not targeted to any clinical symptom or disease; we could only conclude that PA has a more significant physiological impact on local blood flow change than OA, which might be related to the higher stimulation intensity and enhanced performance. To establish the relation of the effect on the blood flow change with the stimulation intensity and the clinical performance, further study simultaneously investigating the clinical effects of PA is needed. In addition, research on the specific mechanisms designed to explain why PA had different physiological influences on local blood flow, e.g. vasoactive substances, are necessary. Moreover, studies using multiple acupoints of various anatomical locations considering possible interactions with each other would be certainly needed based on this study results with one local acupoint. Studies investigating other physiological parameters may be meaningful as future endeavors.

## Conclusions

The study results suggest the possibility of PA in clinical use and its applicability in further investigations, as PA increased local blood flow without any associated discomfort or pain.

## Supporting information

**S1 Fig. Effects of acupunctured sides and sequence on the changes of skin blood perfusion.** The changes of perfusion unit between the right and left legs from baseline for each session were compared by paired t-test. There were no significant differences between right and left blood perfusion in either the PA (**a**) or OA (**b**) group. The changes of perfusion unit by order from baseline for each session were compared by paired t-test. There were no significant differences in the PA (**c**) or OA (**d**) group. Values presented as means ± standard deviations. OA,

ordinary acupuncture; PA, porous acupuncture.
(TIF)

**S1 File. Supplementary data.** Minimal data set.
(XLSX)

**S1 Table. The sensation of acupuncture.** This table was offered to participants to choose one or more sensations they felt during the experiment by each phase.
(DOCX)

**S2 Table. The comparisons in the changes of mean skin blood perfusion between the first and the second sessions.** The t-test was performed to compare the differences between the first and the second sessions. Data represent Mean ± standard deviation. Min, minutes.
(DOCX)

## Author Contributions

**Conceptualization:** Sanhwa Hong, Hong Soo Kim, Yu-Kang Kim, Ju-Young Oh, Hyangsook Lee, Su-Il In, Hi-Joon Park.

**Formal analysis:** Taehun Kim, Jaeha Ahn, Hyun-Seo Song, Ju-Young Oh, Tae-Yeon Hwang.

**Funding acquisition:** Su-Il In, Hi-Joon Park.

**Investigation:** Koh-Woon Kim, Sanhwa Hong, Taehun Kim, Jaeha Ahn, Hyun-Seo Song, Yu-Kang Kim.

**Methodology:** Sanhwa Hong, Hong Soo Kim, Tae-Yeon Hwang, Hyangsook Lee.

**Resources:** Hong Soo Kim, Su-Il In.

**Supervision:** Koh-Woon Kim, Hyangsook Lee, Su-Il In, Hi-Joon Park.

**Writing – original draft:** Koh-Woon Kim, Sanhwa Hong, Hi-Joon Park.

**Writing – review & editing:** Koh-Woon Kim, Hong Soo Kim, Taehun Kim, Jaeha Ahn, Hyun-Seo Song, Yu-Kang Kim, Ju-Young Oh, Tae-Yeon Hwang, Hyangsook Lee, Su-Il In, Hi-Joon Park.

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
