## [Decision Letter · Decision Letter 0]

25 Sep 2019

PONE-D-19-18282

Physiological impact of nanoporous acupuncture needles: laser Doppler perfusion imaging in healthy volunteers

PLOS ONE

Dear Dr Koh-Woon Kim,

Thank you for submitting your manuscript to PLOS ONE. After careful consideration, we feel that it has merit but does not fully meet PLOS ONE’s publication criteria as it currently stands. Therefore, we invite you to submit a revised version of the manuscript that addresses the points raised during the review process.

Thank you for your submission to this journal.

Please find the reviewers comments. Please address each of these points and indicate clearly your response to each and where it can be found in the revised manuscript.

I would like to see more defence of the design. Specifically, 1) the choice of a randomised crossover trial and it be clearly marked on the results tables the data from each group before and after the cross-over 2) defence of the choice of the blinding question to participants 3) consideration in the discussion as to the statistical power of the study (for a large, medium or small effect) and whether the study is powered to find the change that occurred.

One reviewer is quite critical about the choice of one acupuncture point and healthy volunteers therefore please defend these choices heavily in your revised paper.

The other reviewer has made comment about some of your procedural choices and possible effects on microcirculation - please consider these carefully.

I look forward to seeing your revised paper and hope you find the reviewers comments constructively helpful.

Leica

We would appreciate receiving your revised manuscript by Nov 08 2019 11:59PM. To enhance the reproducibility of your results, we recommend that if applicable you deposit your laboratory protocols in protocols.io, where a protocol can be assigned its own identifier (DOI) such that it can be cited independently in the future. For instructions see: http://journals.plos.org/plosone/s/submission-guidelines#loc-laboratory-protocols

We look forward to receiving your revised manuscript.

Kind regards,

Leica S. Claydon-Mueller

Academic Editor

PLOS ONE

Journal Requirements:

1. We note that you have indicated that data from this study are available upon request. PLOS only allows data to be available upon request if there are legal or ethical restrictions on sharing data publicly. For information on unacceptable data access restrictions, please see http://journals.plos.org/plosone/s/data-availability#loc-unacceptable-data-access-restrictions.

Reviewers' comments:

Reviewer's Responses to Questions

**Comments to the Author**

1. Is the manuscript technically sound, and do the data support the conclusions?

Reviewer #1: Partly

Reviewer #2: No

2. Has the statistical analysis been performed appropriately and rigorously? 

Reviewer #1: I Don't Know

Reviewer #2: N/A

3. Have the authors made all data underlying the findings in their manuscript fully available?

Reviewer #1: No

Reviewer #2: Yes

4. Is the manuscript presented in an intelligible fashion and written in standard English?

Reviewer #1: Yes

Reviewer #2: Yes

5. Review Comments to the Author

Reviewer #1: I thought the authors present an excellent piece of research on nanoporous acupuncture needles - there's definitely novelty and implications for use in clinical practice. However, I reckon this manuscript needs minor to major revisions with regard to (I) English language for clarity and style consistent with the standards of scientific / academic rigour (II) inadequate reporting of data from which inferences were drawn.

I have tried to highlight some of the points

Lines 36-37 & 108-109: "In a randomized, controlled crossover trial" - is this a randomised crossover laboratory study?

Line 43: "consequently" - or subsequently?

Lines 91-95: consider rephrasing the sentence - it appears too long and difficult to comprehend.

Line 101: "led to the enhanced therapeutic effect of acupuncture" - therapeutic effect appears vague, is it increased flow of blood in the microcirculation, pain-relief? consider for clarity?

Line 117: "Participants" - I think they are still volunteers at this stage. Once they enroll after signing consent, they become participants.

Lines 117-121: I was wondering what is the reason for your eligibility criteria? Some references might help

Lines 122-125: Consider rephrasing the sentence - unclear

Line 129: consider deleting "trial"

Line 133 & 152-156: "received PA on the left leg and, after 30 minutes of rest, received OA on the right leg" - I think I get the sense of it but some readers might find it difficult to follow - for e.g., was Doppler imaging done during these 30 mins? I think it would be easier to follow if there's a graph showing sequence of events

Lines 141-145: "history of alcohol ingestion, smoking, and medication" - I appreciate the information collected, think these could potentially have affected blood microcirculation? If these were not controlled, I'd consider making a note of it in the limitations section.

Line 155: "consequently" or subsequently?

Line 166: could cleaning with alcohol have altered blood microcirculation?

Lines commencing 359: "study`" single inverted comma might have inadvertently been put?

Other points

1. Statistical analysis seem to have been performed appropriately however reporting data tables might be useful.

2. Other information missing such as was constraint randomisation used? What are the p values in Tables 2 & 3

3. Although the authors have provided conclusion in the abstract, I think it is missing from the main text.

Reviewer #2: This study is a physiological assessment of one new kind of acupuncture (nanoporous acupuncture) compared with traditional acupuncture. Although, the study is innovative, the design of the study is not rigorous. Firstly, the choose of healthy people is not enough since most of the subjects who seek acupuncture treatment is not healthy. Secondly, only one acupoint-ST36 for the assessment is not consistent with the real situation. Usually, we will use more than 10 acupoints per treatment and their blood flow may interact with each other. Thirdly, the effectiveness of this kind of acupuncture is totally unknown and the underlying mechanism of acupuncture is complex and only test blood flow is too far from the traditional acupuncture.

6. PLOS authors have the option to publish the peer review history of their article (what does this mean?). If published, this will include your full peer review and any attached files.

Reviewer #1: Yes: Gourav Banerjee, PhD

Reviewer #2: No

---

## [Author Response · Author response to Decision Letter 0]

29 Oct 2019

Please refer to the attached file, "Cover letter" for response to editor comments and "Response to Reviewers" for response to the reviewers.

---

## [Editor Report · Decision Letter 1]

7 Nov 2019

PONE-D-19-18282R1

Physiological impact of nanoporous acupuncture needles: laser Doppler perfusion imaging in healthy volunteers

PLOS ONE

Dear Dr Koh-Woon Kim,

Thank you for submitting your manuscript to PLOS ONE. After careful consideration, we feel that it has merit but does not fully meet PLOS ONE’s publication criteria as it currently stands. Therefore, we invite you to submit a revised version of the manuscript that addresses the points raised during the review process.

Thank you for your very clear and appropriate responses to my comments and the reviewers.

I have one final suggestion in relation to blinding. Please could some consideration be made, not only about the participants, but the assessors and statistician blinding. For example, did the statistician know which groups received what stimulation when performing the analysis?

Thank you very much.

We would appreciate receiving your revised manuscript by 5 November 2019. To enhance the reproducibility of your results, we recommend that if applicable you deposit your laboratory protocols in protocols.io, where a protocol can be assigned its own identifier (DOI) such that it can be cited independently in the future. For instructions see: http://journals.plos.org/plosone/s/submission-guidelines#loc-laboratory-protocols

We look forward to receiving your revised manuscript.

Kind regards,

Dr. Leica S. Claydon-Mueller

Academic Editor

PLOS ONE

---

## [Author Response · Author response to Decision Letter 1]

11 Nov 2019

My colleagues and I greatly appreciate the editor's final suggestion in relation to blinding and have incorporated the suggestion into the manuscript. 

As there were no additional comments from the reviewers, we omitted “Response to Reviewers” at this time.

Please refer to the "Cover letter" file.

---

## [Editor Report · Decision Letter 2]

25 Nov 2019

Physiological impact of nanoporous acupuncture needles: laser Doppler perfusion imaging in healthy volunteers

PONE-D-19-18282R2

Dear Dr. Koh-Woon Kim,

We are pleased to inform you that your manuscript has been judged scientifically suitable for publication and will be formally accepted for publication once it complies with all outstanding technical requirements.

With kind regards,

Leica S. Claydon-Mueller

Academic Editor

PLOS ONE

---

## [Editor Report · Acceptance letter]

2 Dec 2019

PONE-D-19-18282R2 

Physiological impact of nanoporous acupuncture needles: laser Doppler perfusion imaging in healthy volunteers 

Dear Dr. Kim:

I am pleased to inform you that your manuscript has been deemed suitable for publication in PLOS ONE. Congratulations! Your manuscript is now with our production department. 

With kind regards,

on behalf of

Dr. Leica S. Claydon-Mueller 

Academic Editor

PLOS ONE